# Classifying Young Children with Attention-Deficit/Hyperactivity Disorder Based on Child, Parent, and Family Characteristics: A Cross-Validation Study

**DOI:** 10.3390/ijerph19159195

**Published:** 2022-07-27

**Authors:** Evelyn Law, Georgios Sideridis, Ghadah Alkhadim, Jenna Snyder, Margaret Sheridan

**Affiliations:** 1Yong Loo Lin School of Medicine, National University of Singapore, Singapore 119077, Singapore; evelyn_law@nuhs.edu.sg; 2Department of Paediatrics, Khoo Teck Puat-National University Children’s Medical Institute, National University Health System, Singapore 119228, Singapore; 3Singapore Institute of Clinical Sciences, Agency of Science, Technology and Research, Singapore 117609, Singapore; 4ICCTR, Boston Children’s Hospital, Harvard Medical School, Boston, MA 02115, USA; 5Department of Primary Education, National and Kapodistrian University of Athens, 157 72 Athens, Greece; 6Department of Psychology, College of Arts, Taif University, P.O. Box 11099, Taif 21944, Saudi Arabia; ghadah.s@tu.edu.sa; 7Department of Psychology and Neuroscience, University of North Carolina, Chapel Hill, NC 27599, USA; jmsnyder27@gmail.com (J.S.); sheridan.margaret@unc.edu (M.S.)

**Keywords:** attention-deficit/hyperactivity disorder, SES, preschool

## Abstract

We aimed to identify subgroups of young children with differential risks for ADHD, and cross-validate these subgroups with an independent sample of children. All children in Study 1 (N = 120) underwent psychological assessments and were diagnosed with ADHD before age 7. Latent class analysis (LCA) classified children into risk subgroups. Study 2 (N = 168) included an independent sample of children under age 7. A predictive model from Study 1 was applied to Study 2. The latent class analyses in Study 1 indicated preference of a 3-class solution (BIC = 3807.70, *p* < 0.001). Maternal education, income-to-needs ratio, and family history of psychopathology, defined class membership more strongly than child factors. An almost identical LCA structure from Study 1 was replicated in Study 2 (BIC = 5108.01, *p* < 0.001). Indices of sensitivity (0.913, 95% C.I. 0.814–0.964) and specificity (0.788, 95% C.I. 0.692–0.861) were high across studies. It is concluded that the classifications represent valid combinations of child, parent, and family characteristics that are predictive of ADHD in young children.

## 1. Introduction

Attention-Deficit/Hyperactivity Disorder (ADHD) is a prevalent neurobiological disorder characterized by inattention, hyperactivity and/or impulsivity. Children diagnosed with ADHD have severe negative outcomes compared to children without ADHD in terms of physical and mental health comorbidity, family and peer difficulties, substance use, academic problems, mortality rates and psychosocial adversity [1,2,3,4,5,6,7,8]. The majority of studies on evidence-based ADHD treatments indicate beneficial outcomes with greater effects near the time of treatment [9,10,11,12]. Evidence-based parent training and behavioral therapy programs reduce disruptive behaviors and ADHD symptoms, and improve parental competence and parent-child interactions [13,14,15]. While these studies give impetus to treat very young children with ADHD, diagnosis at a young age is complicated by the normal developmental, behavioral, and temperament variations during early years [15,16].

The diagnostic stability of ADHD diagnosis from preschool to middle childhood within community samples is approximately 50% [17,18,19,20,21]. However, studies using more rigorous criteria for ADHD, many of which include comprehensive psychological evaluation, have shown a higher stability of diagnosis from 65 to 89.2% across this time interval [22,23,24,25,26]. The Preschool ADHD Treatment Study (PATS) followed preschoolers (mean age of 4.4 years at initial evaluation) with moderate-to-severe symptoms and high level of global impairment. Of these children, 89.2% continued to meet criteria for ADHD diagnosis after 6 years [26]. Finding such as these raise the possibility that some combination of child characteristics measured in preschool could predict which children were likely to continue to meet criteria for ADHD in middle childhood and which were not. Thus far, preschool studies have found that the presence of comorbid disruptive disorders, later age at initial assessment, and poorer performance on preschool measures of behavioral inhibition increase risk for ADHD persistence [21,26,27,28].

While these studies have examined predictors of ADHD diagnostic stability, they have relied on testing a priori hypotheses. In contrast, person-centered analytic approaches are able to model clinical population heterogeneity to identify novel subgroups of children who are at disproportionately higher risk for more or less favorable outcomes. Thus, one way to approach this is to use person-centered or latent class analysis, which is a non-parametric variant of cluster analysis that postulates a discrete latent variable to classify individuals with similar symptoms into more homogenous subgroups. This kind of analysis has been used to identify subtypes of ADHD in the past, specifically classifying children based on parent-reported symptoms, child/adolescent reported symptoms, personality, comorbidity, neuropsychological profiles, and trajectory of symptoms [9,29,30,31,32,33,34,35,36,37,38,39,40,41,42]. While most of these studies have identified subgroups in older children and adolescents with ADHD, only a few LCA studies are completed in preschool children [43,44]. Studies using the LCA approach in preschoolers have demonstrated empirically formed ADHD subgroups of preschoolers with different neuropsychological impairments; however, these subgroups do not show differences in their persistence of ADHD symptoms. The authors conclude that while neuropsychological impairments can accurately differentiate inattentive and hyperactive young children from typically developing children, their neuropsychological performance does not predict ADHD status later because ADHD trajectories are likely influenced by an array of subsequent environmental and neurodevelopmental factors.

In a most relevant recent study [22] the authors examined a combination of environmental and behavioral correlates in predicting ADHD persistence using latent class analysis through following 120 young children (mean age 5.7 years) diagnosed with ADHD after multidisciplinary assessments with psychologists and pediatricians in a tertiary care program. After a mean of 7 years (range: 5.6–9.5 years), 73.3% of the cohort were re-evaluated for ADHD and of those 70.4% continued to meet research diagnostic criteria for ADHD. Predictors of diagnostic stability were elevated externalizing and internalizing behaviors at the time of ADHD assessment, parental history of psychopathology, and the family’s socioeconomic status (SES) as measured by the income-to-needs ratio. Using latent class analysis, they found three subgroups of children that shared similar characteristics with each other. Two subgroups described children who had persistent ADHD over time and one described children who later did not meet diagnostic criteria for ADHD. The first persistent ADHD subgroup consisted of children who came from families with low SES. The second subgroup consisted of children who had mothers with high educational level and parents had high level of psychopathology. The third subgroup represented children with remitted ADHD, characterized by lower externalizing and internalizing symptoms at baseline and high SES parents.

The above study profiled individuals with ADHD to identify the presence of meaningful subgroups of individuals and found predictive validity for ADHD stability using these subgroups. While this study was an important first step in identifying pathways through which ADHD symptoms became persistent from early to middle childhood, it was possible that classes reflected idiosyncratic sample-specific patterns. Since latent class analysis was a data-driven approach to identifying subgroups, sample characteristics could have a profound effect on the groups identified. As the previous study was on an ADHD sample from a tertiary care clinic, it might also represent children with more severe impairments from ADHD and not the entire population of children with ADHD during early childhood. Importantly, the relevance of these subgroups to the wider population of individuals with ADHD could only be determined through replication.

In Study 1, we examined the percentage of ADHD diagnostic stability from preschool to adolescence in a tertiary care program that diagnosed each child’s preschool ADHD diagnosis with neuropsychological testing. In Study 2, we utilized more comprehensive parent and family socioeconomic variables to establish subgroups of children with differential ADHD risk and cross-validate the presence of these clusters in a second, unrelated sample of preschool children from the community.

## 2. Methods

### 2.1. Participants

#### 2.1.1. Study 1

Participants were 120 children (21 girls and 99 boys) between the ages of 3-years, 0-month and 6-years, 11-months with a mean age of 5-years, 7-months (S.D. = 11.35 months) from the Boston metropolitan area in the USA. These children underwent assessments by a multidisciplinary team comprised of pediatric psychologists and developmental-behavioral pediatricians and were consecutively diagnosed with ADHD from 2003 to 2008.

After a mean follow-up interval of 7 years later, confirmation of ADHD was conducted using the Diagnostic Interview Schedule for Children-4th Edition (DISC-IV), a structured parent interview, the Parent and Teacher Vanderbilt ADHD Rating Scales, and the Parent Global Assessment scale (PGA), a 7-point scale ranging from 1 (no impairments) to 7 (needs 24-h supervision due to severe impairments) [45,46]. ADHD diagnosis was considered present if children met criteria for ADHD on the DISC-IV or met intermediate criteria for ADHD on the DISC-IV, had >6 inattention and/or hyperactivity/impulsivity on the Vanderbilt ADHD Diagnostic Parent and Teacher Rating Scales, showed impairments in multiple settings, and had a PGA score > 3 (i.e., moderate to severe impairments). For further details, see [22].

#### 2.1.2. Study 2

Participants were 168 children (54 girls and 114 boys) between the ages of 3-years, 0-month and 6-years, 11-months with a mean age of 5-years, 8-months (S.D. = 14.51 months) again recruited from the Boston metropolitan area in the USA. Eighty-four children were diagnosed ADHD using the DISC-IV (Young Children Version), Child Behavior Checklist (CBCL) [47], and the Swanson, Nolan, and Pelham Rating Scale (SNAP-IV) [48]. ADHD criteria were met if they had sufficient symptoms of ADHD and reported impairment on the DISC-IV or if they met criteria for an intermediate diagnosis on the DISC-IV, had a T-score greater than 70 on the CBCL Externalizing Scale, and had >6 symptoms of inattention and/or >6 symptoms of hyperactivity/impulsivity endorsed on the SNAP-IV. These 84 children were matched by age to 84 typically developing children without ADHD. Both studies were approved by the respective Institutional Review Boards.

### 2.2. Measures

#### Predictors of ADHD

Table 1 described child, parent, and family characteristics used in the latent class analysis for both Study 1 and 2. We used severity of externalizing symptoms, severity of internalizing symptoms, parental history of psychopathology, income-to-needs ratio, maternal education level, household income, and neighborhood poverty level as potential subgroup predictors in our latent class analysis to determine whether the socioeconomic environment shaped risks for ADHD.

In Study 1 and 2, we measured the same constructs and included these identical constructs in the latent class analysis. Replication of our observations from Study 1 in an independent sample (Study 2) constituted a full and independent replication of our observations. In addition to directly replicating our findings in two samples, we also tested the predictive validity of Study 1 findings through developing a set of predictive equations regarding latent class membership and testing these equations with Study 2 data. For example, we tested whether the children with and without ADHD from Study 2 were aligned correctly to their latent classes using predictive equations developed in Study 1. This allowed us to determine whether equations developed in a sample (Study 1) could be used to obtain predicted membership probabilities for a second sample (Study 2).

### 2.3. Statistical Data Analyses

#### 2.3.1. Study 1 Latent Class Analysis

In performing a latent class analysis, one concern is identifying the optimum numbers of clusters. We address this using both theoretical and methodological means [53]. Theoretically speaking, parsimony and meaningfulness are the prevailing attributes of an optimum latent class structure, we describe the way in which our observed subgroups meet these criteria below [54]. In addition, we report on several criteria which indicate the appropriate nature of the observed classes: (a) high entropy values based on correct classifications, (b) low misclassifications, (c) significance of predictors in defining latent class membership, (d) significant reduction in error variances when moving from one-latent class solution to the next, (e) large amounts of standard explanatory variance, (f) adequate group sizes per latent class, (g) small Akaike Information Criterion (AIC) and Bayesian Information Criterion (BIC) values as more classes are estimated, and, (h) superiority of one model over the other using the likelihood ratio test and through employing the bootstrap distribution with a proper number of replications (e.g., 1000). All analyses were conducted using the Latent Gold 5.0 software [55].

#### 2.3.2. Study 2 Latent Class Mixture and Predictive Modeling

The analytic strategy employed in Study 1 was replicated in Study 2 with one important addition [56]. Besides estimating a new latent class model using the Study 2 data and visually examining its similarity with the Study 1 solution, a predictive model from Study 1’s latent class solution was estimated and applied to the data from Study 2. This approach was taken to determine how many of the children in Study 2 would be properly assigned to latent classes created using Study 1 data. Indices of sensitivity, specificity, positive and negative predictive value and likelihood ratio tests were estimated for the goodness with which Study 1 latent class structure classified Study 2 participants [57]. Here, a true positive would be a participant in Study 2 diagnosed with ADHD who is predicted to belong to Latent Class 1, using Study 1’s latent class model.

#### 2.3.3. Power Analysis for LCA

Our proposed model was simulated for the presence of 3 latent classes and membership coefficients in logits ranging between −3 and +1 distances. Using 1000 replications of sample sizes equal to 100 participants results indicated that entropy values (accuracy of classifications based on model parameters) were equal to 95.1%. The model was slightly overpowered with regard to the Likelihood Ratio Chi-square test (95.9% rejections compared to 95% defined by the nominal alpha level) with estimates of AIC and BIC being also very close to 95% (means of 95.4 and 95.3% for AIC and BIC, respectively). These estimates, however, can be markedly different compared to observed estimates, for which proper values were unavailable given absence in the literature of relevant studies.

## 3. Results

### 3.1. Prerequesite Analyses: Parental Psychopathology and the Effecst of Gender

Data on specific conditions related to parental psychopathology were available from Study 1 data only. These results are presented here for descriptive purposes only as sample size restricted further predictions using specific conditions of parental psychopathology. Nevertheless, in Study 1, there was a high prevalence of parental history of ADHD (53.3%), anxiety/depression (43.9%), a learning disability (LD, 42%), and a bipolar disorder (25.8%).

For gender differences inferential statistics contrasted boys and girls on either level (means) or frequency of a categorical variable (e.g., internalizing problems). Results pointed consistently to the absence of gender differences. Specifically, there were no differences in levels of poverty [Study 1: *F*(1, 119) = 0.274, *p* = 0.602]; Study 2: *F*(1, 153) = 0.828, *p* = 0.364], Income-to-Needs Ratio [Study 1: *F*(1, 84) = 2.560, *p* = 0.113; Study 2: *F*(1, 144) = 0.741, *p* = 0.391], salary [Study 1: *F*(1, 86) = 2.002, *p* = 0.161; Study 2: *F*(1, 148) = 0.283, *p* = 0.596], parental psychopathology [Study 1: χ^2^(1) = 0.216, *p* = 0.642; Study 2 χ^2^(1) = 0.007, *p* = 0.933], externalizing symptoms [Study 1: χ^2^(4) = 4.430, *p* = 0.351; Study 2 χ^2^(1) = 1.986, *p* = 0.159], internalizing symptoms [Study 1: χ^2^(4) = 2.192, *p* = 0.700; Study 2 χ^2^(1) = 0.043, *p* = 0.836], and parental education [Study 1: χ^2^(1) = 0.014, *p* = 0.906; Study 2 χ^2^(1) = 3.114, *p* = 0.078].

### 3.2. Latent Class Modeling for ADHD Data

Initially a 1-class latent mixture model was applied to the data in order to provide a baseline on which other models were compared, as this 1-class model did not have the presence of subgroups. The first comparison involved the fit from a 2-class to the baseline model. When the combination of child, parent, and family variables were modeled, results indicated that the 2-class structure fit the data from Study 1 better compared to a 1-class solution (Table 2). Indices of BIC were lower, and classification errors were non-significant. The second comparison involved the fit of a 3-class model to the data. Results suggested superior fit of the 3-class model over the 2-class model and that finding was confirmed from the Likelihood Ratio difference test using the Bootstrap distribution (LL = 190.297, *p* < 0.001) and the descriptive fit indices (BIC_2-class_ = 3935.638, BIC_3-class_ = 3807.702). The number of children per class were 38 (Class 1: High ADHD probability), 35 (Class 2: Partial/Intermediate ADHD probability), and 13 (Class 3: Very low ADHD probability). Classification errors were equal to 0.0042%. Given those numbers it was considered reasonable to not pursue additional classes, as the number of cases would significantly reduce and would likely invalidate the generality of the emerged solution. Figure 1 displayed the obtained 3-class solution. Class 1 was the “High ADHD probability/low SES” class; 94.5% of the children in this class with an ADHD diagnosis at baseline and follow-up had on average three of the indices of SES (neighborhood poverty, family income, and income-to-needs ratio) that were lower than in the other two classes of children. Further, both externalizing symptoms and internalizing symptoms were significantly elevated, compared to the other two classes. The findings of Study 1 suggested the presence of a ‘disadvantaged’ profile for children with ADHD that originated in multiple sources: the person (increased psychopathology) and the home environment (low SES).

### 3.3. Validation of Latent Class Structure of Study 1 Data Using Study 2 Data

As shown in Table 2, the 3-class solution was again the observed choice of model fit in Study 2. The 3-class model was associated with the lowest BIC values. Second, indices of entropy and standard R-square values were high (R_Standard_^2^ = 99%; R_entropy_^2^ = 98%). Figure 2 displayed the latent class structure observed using Study 2 data. The observed latent class solution was identical to that emerged using Study 1 data. Few quantitative findings differentiated the two solutions. For example, parent history of psychopathology and children’s severity of internalizing symptoms were no longer significant predictors of the high ADHD probability/low SES latent class but approached significance (see Table 3). All other findings pointed to the presence of very similar latent class solutions. Thus, the role of the child, parent, and family factors to the existence of ADHD was largely identical between two independent groups of students with ADHD. The next section described the findings from applying the predictive model of Study 1 to Study 2 data.

### 3.4. Latent Class Predicted Membership at Time 2 with Time Predictive Equations

This modeling approach utilized information from the latent class analysis of Study 1 to develop a series of predictive equations from that dataset. We then applied those predictive equations to the data in Study 2 and evaluated how well the known membership was aligned with those from the predictive model. Agreement between predicted and actual membership scores regarding ADHD and non-ADHD membership was tested using sensitivity and specificity analysis. As shown in Table 4, indices of sensitivity and specificity were high and by all means acceptable using conventional standards [56,58]. Thus, the probability that a child with ADHD in Study 2 data would be classified into the “High ADHD probability/low SES” class of Study 1 data (latent class 1) was 91.3%. Similarly, the probability that a child without ADHD would belong to any of the remaining two latent classes was 78.9%, suggesting the presence of some misclassified cases. Additionally, true positive and true negative rates were 75% and 92.8%, respectively. Furthermore, indices of positive and negative predictive power controlled for prevalence rates were similarly high i.e., 75% and 93%, respectively.

## 4. Discussion

The purpose of the present studies was to identify and confirm the presence of subgroups of young children with ADHD as a function of child, parent, and family characteristics. We found an identical three-class solution in Study 1 and the data were replicated using an independent sample of children in Study 2. In both studies, we observed that three groups of children with ADHD emerged-one group who was socioeconomically advantaged and had a low likelihood of carrying an ADHD diagnosis and two groups of more disadvantaged youths who were more likely to meet criteria for ADHD diagnosis.

The present studies were designed to add to the extant literature in a number of ways. To our knowledge, previous studies examined how child-specific predictors, such as neuropsychological deficits, group together within individuals, and did not include parent and family predictors. This was the first person-centered analysis to investigate how child, parent, and family variables combined across individuals. This analysis differentiated young children with and without ADHD into subgroups, and these subgroups were cross validated in an independent sample with high sensitivity and specificity.

Importantly, this study found that parent and family factors defined class membership more strongly than child factors. Maternal education, household income, and income-to-needs ratio were the most discriminating variables in the prediction of ADHD in young children with R^2^ = 0.512, 0.886, and 0.831, respectively, indicating that children who came from disadvantaged backgrounds tended to belong in the classes with higher probabilities of ADHD.

It has been well documented that parents who are under-resourced have higher rates of adversity exposures, increased exposure to stress, and lower parental involvement, all of which are likely to increase risks for psychopathology [60,61,62,63,64,65,66]. In particular, an emerging literature associating low SES and other forms of adversity exposure with poor executive functions (EF) and high risk for ADHD have led to a theoretical model of the impact of early experience on EF and externalizing psychopathology [64,65,67]. Potential mediators of the relationships between low SES and high risk for ADHD include the quality of the home environment (e.g., amount of EF practice, use of complex language, enrichment activities, and quality of parent-child interactions) [68]. Finally, access to intervention may play a role in explaining our observations in this study. Parents with ample resources may have good access to early intervention strategies and resources to implement them. They are more likely to provide a day-to-day environment for children which scaffolds executive function (EF) skills and hence child symptoms of ADHD observed in preschool improve by middle childhood [22].

Our findings supported the complex interplay of both child and family factors in the susceptibility of ADHD. While twin, family, and adoption studies have demonstrated a strong heredity component in ADHD symptoms with a twin heritability estimate of ADHD of 0.77 [36,40,69], more recent studies have examined environmental factors in susceptibility of this condition [70,71]. Consistent with this literature [60,61,63,66,67,68,69,70,72,73,74,75,76], we observe that both severity of early ADHD symptoms and family socioeconomic background are important in differentiating those with varying probabilities of ADHD diagnosis in preschool.

In summary, clinicians should consider family SES when recommending and implementing treatment for young children with ADHD, as children in low SES families may be the most vulnerable. When family risk factors are present, clinicians should consider interventions that target the home environment as well as the child’s specific behaviors, (e.g., parent training programs, along with school accommodations for the child).

The present study was limited for several reasons. First, the sample size of Study 1 was modest and would allow for idiosyncrasies in the observed latent class membership. However, the stability with which the latent class structure observed in both Study 1 and 2 data was very high. Furthermore, the application of the predictive model of Study 1 latent structure with Study 2 data was associated with high rates of accurate classification of ADHD cases and non-ADHD cases. Thus, adoption of the present methodology and the use of cross-validation remarkably lowered the possibility that the present findings were idiosyncratic and non-representative of the population.

With a modest sample size, this study did not include all environmental exposures in early childhood, including measures of violence exposure, prenatal exposures, nutrition, psychosocial supports of families, and parental stress. Additionally, this study was unable to disentangle many interrelated variables such as SES and increased genetic predisposition. The variable of “parental history of psychopathology” captured genetic disposition of ADHD of a child, but at the same time, could be an indicator of suboptimal parenting. These interactions were difficult to tease out due to the relatively small sample size where a large number of parent psychopathology predictors would be suboptimal from the statistical point of view. However, anecdotally, there was a high prevalence of anxiety/depression, ADHD, and a learning disability in the parents of the children. Furthermore, our coding scheme in Study 2 did not specifically record each and every one of parent psychopathologies but instead utilized a binary system for the presence of any psychopathology. Certainly, a larger scale, prospective study is needed to further pursue these goals.

In the future, this study may be extended to include a follow-up of children in Study 2, in order to identify and cross-validate subgroups of children who are at the highest probability of persistent ADHD over time. This kind of LCA will have utility in predicting future trajectories of individuals with ADHD and will allow clinicians to decipher those who are at greatest risk of ADHD persistence from early to middle childhood, ensuring that the limited resources available for intervention are provided for the children most in need of them.

## 5. Conclusions

In conclusion, this study points to the important environmental and home influences on the susceptibility of ADHD. To further expand our understanding of ADHD heterogeneity and its current classification, employment of additional variables related to genetic, physiological, academic neurocognitive measures, and neuroimaging findings are warranted in future studies.

## Figures and Tables

**Figure 1 ijerph-19-09195-f001:**
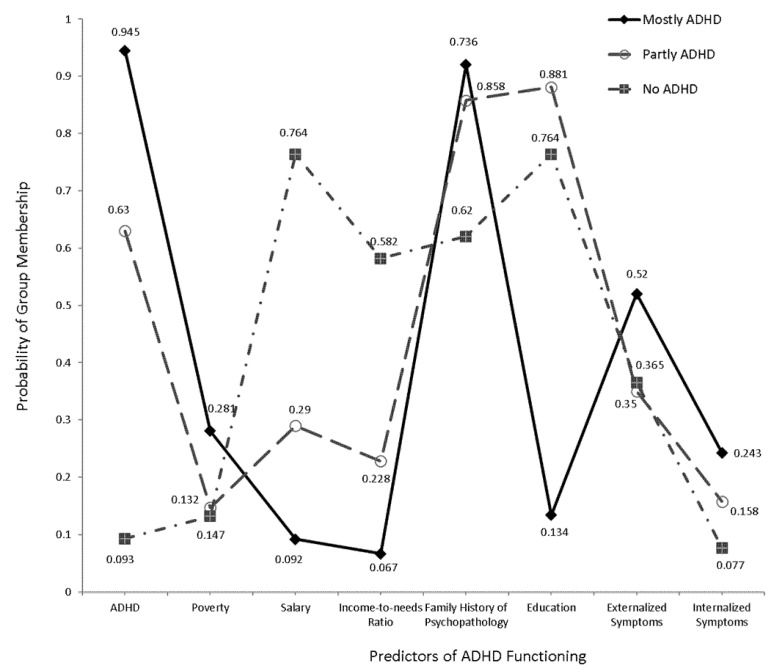
Three-Class latent class model using Study 1 data.

**Figure 2 ijerph-19-09195-f002:**
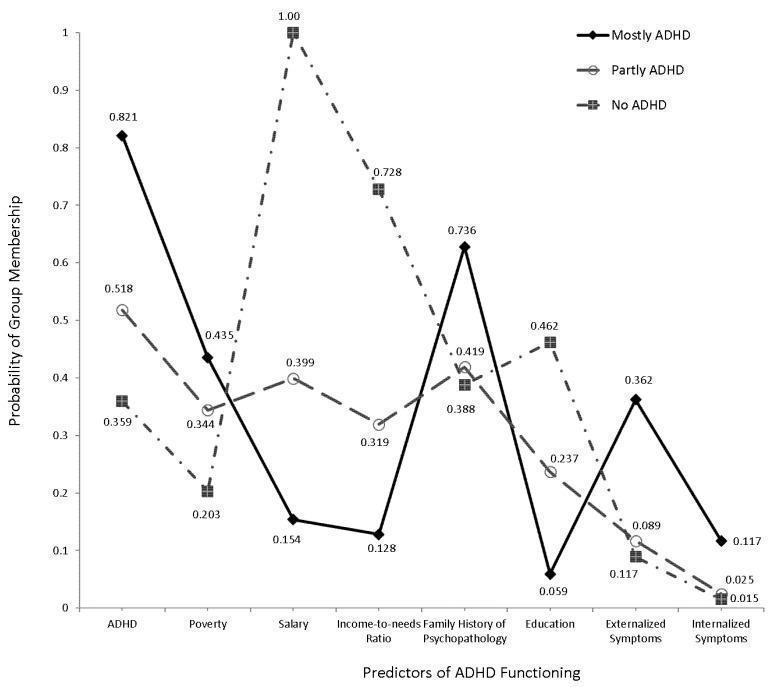
Three-Class latent class model using Study 2 data.

**Table 1 ijerph-19-09195-t001:** Child, Parent, and Family Characteristics.

Level	Characteristic	Definition
Child	IQ	Composite nonverbal cognitive score on the Wechsler Preschool and Primary Scales of Intelligence, 3rd edition [49]
Child	Severity of externalizing symptoms	Obtained from the Behavior Assessment System for Children (BASC-2) [50]; and Child Behavior Checklist (CBCL) [47] stratified to T-score ≥ or <70
Child	Severity of internalizing symptoms	BASC-2 and CBCL, stratified to T-score ≥ or <70
Child	Age at diagnosis	Age at the time of psychology assessment in months
Child	Gender	Male or female by parent report
Parent	Parental history of psychopathology	Obtained from the Family Interview for Genetic Studies [51]. Examples of conditions observed were anxiety/depression, bipolar disorder, ADHD, and learning disabilities
Parent/Family	Socioeconomic status (SES): Maternal education	By parent report on the MacArthur SES questionnaire [52]. Stratified into high school or less, some college, or at least a college degree
Family	SES: Neighborhood poverty	Percentage of residents living below the poverty level based on the family’s address according to the US census
Family	SES: Income	Total family yearly income by parent report
Family	SES: Income-to-needs	Income divided by the poverty line by family size from the US Census Bureau

**Table 2 ijerph-19-09195-t002:** Nested Latent Class Models Suggesting the Superiority of a 3-Class Solution over Competing 1-, 2-class models for Modeling ADHD Subgroups with Child, Parent, and Family Predictors.

Model	LL	BIC (LL)	# Parameters	Classification Error
Study 1 Data				
1-Class	−2024.496	4106.899	13	<0.001
2-Classes	−1907.685	3935.638	27	0.053
3-Classes	−1812.537	3807.702	41	0.004
Study 2 Data				
1-Class	−2826.715	5707.223	11	<0.001
2-Classes	−2523.565	5159.608	23	<0.001
3-Classes	−2468.423	5108.008	35	0.019

**Table 3 ijerph-19-09195-t003:** Three-Class Model and Significance of Indicators on Classifying Three Independent Classes of Participants.

Predictors	Class 1	Class 2	Class 3	Wald Test	*p*-Value	R^2^
Study 1						
ADHD	0.945	0.630	0.093	19.169	<0.001 ***	0.394
Poverty	11.329	6.131	5.545	9.589	0.008 **	0.109
Salary	32,173	90,581	230,407	373.336	<0.001 ***	0.886
Income-to-Needs Ratio	1.572	4.748	11.690	304.077	<0.001 ***	0.831
History of Parental Psychopathology	0.920	0.858	0.620	6.018	0.049 *	0.080
Maternal Education	0.134	0.881	0.764	31.921	<0.001 ***	0.512
Externalizing	2.082	1.399	1.461	7.998	0.018 *	0.087
Internalizing	0.973	0.631	0.307	7.271	0.026 *	0.054
Study 2						
ADHD	0.821	0.518	0.359	12.689	0.002 **	0.119
Poverty	17.545	14.165	8.940	21.545	<0.001 ***	0.151
Salary	35,082	82,901	199,999	1918.309	<0.001 ***	0.970
Income-to-Needs Ratio	1.496	3.510	7.804	610.751	<0.001 ***	0.828
History of Parental Psychopathology	0.628	0.419	0.388	3.855	0.150	0.034
Maternal Education	0.059	0.237	0.462	11.301	0.003 **	0.117
Externalizing	0.362	0.117	0.089	9.001	0.011 *	0.085
Internalizing	0.117	0.025	0.015	4.121	0.130	0.042

Note: * *p* < 0.05; ** *p* < 0.01; *** *p* < 0.001.

**Table 4 ijerph-19-09195-t004:** Indices of sensitivity and specificity for agreement between latent classes of Study 1 and 2.

	Estimate	95% Confidence Interval ^†^	
		Lower Limit	Upper Limit
Sensitivity	0.913	0.814	0.964
Specificity	0.788	0.692	0.861
For any case of ADHD identified as belonging to the same class using Study 2 data the probability that it is:			
True Positive is:	0.750	0.642	0.835
False Positive is:	0.250	0.165	0.358
For any particular negative test result, the probability that it is:			
True Negative is:	0.928	0.845	0.970
False Negative is:	0.071	0.029	0.155
Conventional L.R. Positive	4.304	2.924	6.336
Conventional L.R. Negative	0.110	0.051	0.238
Weighted L. R. Positive	3.000	2.030	4.433
Weighted L. R. Negative	0.077	0.035	0.167
Positive Predictive Power (PPP)	0.750	0.644	0.838
Negative Predictive Power (NPP)	0.929	0.851	0.973

Note: The conventional positive likelihood ratio is defined as the ratio of (sensitivity)/(1 − specificity); the conventional negative likelihood ratio is defined as the ratio of (1 − sensitivity)/(specificity); the weighted positive likelihood ratio is defined as the ratio of (prevalence)(sensitivity)/(1 − prevalence)(1 − specificity) to control for the levels of prevalence; the weighted negative likelihood ratio is defined as the ratio of (prevalence)(1 − sensitivity)/(1 − prevalence)(specificity) to control for levels of prevalence. PPV is estimated as follows:$\;\mathrm{PPV}=\frac{\mathrm{Sensitivity}*\;\mathrm{prevalence}}{\mathrm{Sensitivity}*\;\mathrm{Prevalence}+\left(1-\mathrm{Specificity}\right)*\left(1-\mathrm{Prevalence}\right)}$. Additionally, NPV is estimated as follows (Gardner & Greiner, 2006): $\mathrm{NPV}=\frac{\mathrm{Specificity}*\left(1-\mathrm{prevalence}\right)}{\left(1-\mathrm{Sensitivity}\right)*\;\mathrm{Prevalence}+\mathrm{Specificty}*\left(1-\mathrm{Prevalence}\right)}$. ^†^ Estimation of confidence intervals involved the following equation: $p\pm 1.96\sqrt{\frac{p\left(1-p\right)}{n}}$ with *p* being the relevant proportion [59].

## Data Availability

Data can be provided by the first author upon request.

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
