# Peer review of "Classifying Young Children with Attention-Deficit/Hyperactivity Disorder Based on Child, Parent, and Family Characteristics: A Cross-Validation Study"

_ijerph, 2022, doi:10.3390/ijerph19159195_

Round 1

Reviewer 1 Report

Law et al reported a predictive model of young children with ADHD with several risk factors and applied the model to another study. Although the latent class analyses of ADHD with several risk factors has been reported in several papers, this study adds more samples and information and supports the critical role of environment in children’s ADHD. 

It would be great if the author adds detailed information about participants, for example, the country and region where these young children are from and got assessed. So that it would be a good reference for the region-specific diagnostic. 

Are there any differences between boys and girls? 

It would be helpful if the author could list some types of parental psychopathology for “parental history of psychopathology”, at least state which kinds of parental psychopathology are most likely to cause children ADHD.

Author Response

Please see attached responses. thank you!

Reviewer 2 Report

Interesting and novel studies. 

I commend the inclusion of epigenetic variables that intervene the development.

It is important that the readers in general, know that ADHD will vary as a result of the conjunction factors.

Author Response

see attached response to queries, thank you!
